# Dichotomy between in-plane magnetic susceptibility and resistivity anisotropies in extremely strained BaFe$_2$As$_2$

Mingquan He [1], Liran Wang[1], Felix Ahn[2], Frédéric Hardy[1], Thomas Wolf[1], Peter Adelmann[1], Jörg Schmalian[1,3], Ilya Eremin[2,4] & Christoph Meingast[1]

High-temperature superconductivity in the Fe-based materials emerges when the antiferromagnetism of the parent compounds is suppressed by either doping or pressure. Closely connected to the antiferromagnetic state are entangled orbital, lattice, and nematic degrees of freedom, and one of the major goals in this field has been to determine the hierarchy of these interactions. Here we present the direct measurements and the calculations of the in-plane uniform magnetic susceptibility anisotropy of BaFe$_2$As$_2$, which help in determining the above hierarchy. The magnetization measurements are made possible by utilizing a simple method for applying a large symmetry-breaking strain, based on differential thermal expansion. In strong contrast to the large resistivity anisotropy above the antiferromagnetic transition at $T_N$, the anisotropy of the in-plane magnetic susceptibility develops largely below $T_N$. Our results imply that lattice and orbital degrees of freedom play a subdominant role in these materials.

[1] Institute for Solid State Physics, Karlsruhe Institute of Technology, 76021 Karlsruhe, Germany. [2] Institut für Theoretische Physik III, Ruhr-Universität Bochum, D-44801 Bochum, Germany. [3] Institute for Theory of Condensed Matter, Karlsruhe Institute of Technology, 76131 Karlsruhe, Germany. [4] National University of Science and Technology "MISiS", 119049 Moscow, Russia. Mingquan He and Liran Wang contributed equally to this work. Correspondence and requests for materials should be addressed to C.M. (email: christoph.meingast@kit.edu)

One striking similarity between iron-based superconductors and high $T_c$ cuprate superconductors is that superconductivity emerges in close proximity to a magnetic instability[1–3]. Most iron pnictides have a stripe-type antiferromagnetic (AF) phase, in which the Fe magnetic moments are parallel to the ordering wave vector either $\mathbf{Q}_1 = (\pi, 0)$ or $\mathbf{Q}_2 = (0, \pi)$, which breaks the $C_4$ symmetry of the paramagnetic structure[4–7]. The magnetic transition at $T_N$ is accompanied, or sometimes even preceded, by a small orthorhombic structural distortion at $T_S \geq T_N$, which has raised the question of whether magnetism alone is driving these transitions[8–11], or whether orbital degrees of freedom also need to be considered[12–15]. This issue is particularly pressing for FeSe, which has no long-range magnetic order down to the lowest temperature at ambient pressure but nevertheless exhibits a similar orthorhombic distortion as the other Fe-based materials[16–18]. This non-magnetic and orthorhombic phase has been coined "electronic nematic"[10, 19]. Experimentally, the susceptibility to form a nematic state has been probed by a variety of methods, including angle-resolved photoemission spectroscopy[20], elastic[21–24], resistivity anisotropy using a piezo stack[25–28], Raman scattering[29–31], thermopower[32], nuclear magnetic resonance[33, 34], optical conductiviy[35, 36]. Interestingly, many optimally doped Fe-based materials appear to be close to a putative nematic quantum critical point[28], and recent theoretical works suggest that electronic nematic fluctuations may provide a boost to superconductivity in various channels[37].

Here we study the interplay between lattice, orbital, magnetic, and nematic degrees of freedom in the parent compound $BaFe_2As_2$ by measuring the in-plane anisotropies of both the uniform magnetic susceptibility and the resistivity under a large symmetry-breaking strain. Measurements of the anisotropic susceptibility, which in the past were not feasible due to the large detwinning devices, are made possible by using a simple approach.

## Results

**Experimental setup.** Figure 1a presents the schematic of our experimental device. Samples were glued onto a glass-fiber-reinforced plastic (GFRP) substrate with the crystal's tetragonal $[110]_{tet}$ direction orientated parallel to the fibers. Figure 1b shows that the difference of the thermal expansion parallel and perpendicular to the fiber direction of the substrate material is comparable in magnitude to the orthorhombic distortion of a free standing $BaFe_2As_2$ crystal[23, 38] near the transition temperature. Thus, by glueing the $BaFe_2As_2$ crystal to this substrate at room temperature, a large symmetry-breaking strain $|\varepsilon_a - \varepsilon_b| = \left| \Delta L_{||}/L_{||}^{300K} - \Delta L_{\perp}/L_{\perp}^{300K} \right| \sim 4 \times 10^{-3}$ can be expected at 140 K. The extremely soft shear modulus near the transition[22] ensures that the strain is fully transmitted to the crystal, as evidenced by the very large resistivity anisotropy and correspondingly large elastoresistivity (see text below and Fig. 2a, b). As will be shown in the following, our symmetry-breaking straining technique allows us to study the response of both the in-plane resistivity and susceptibility anisotropies under these extreme strain conditions. Magnetization measurements are possible due to the very small size, as well as the fairly weak magnetic response of the substrate.

**In-plane resistivity and susceptibility anisotropies.** The measured in-plane resistances of $BaFe_2As_2$ in the symmetry-breaking straining setup are shown in Fig. 2a. The resistances $R_b$ and $R_a$ were measured on the same sample and are normalized by the resistances at 300 K. In general it is non-trivial to obtain the resistivity anisotropy from the measured resistances in this geometry due to non-uniform current flow[39]. However, for the

present small in-plane and out-of-plane anisotropies[25, 26, 28, 40, 41] the corrections are expected to be small[39], and in the following we assume that $\rho_b/\rho_a \sim R_b/R_a$. Our in-plane resistivity anisotropy with $\rho_b > \rho_a$ is consistent with the largest anisotropy $(\kappa = \rho_b/\rho_a - 1)_{max} \sim 40\%$ obtained by conventional detwinning methods[41–45], proving that the sample experiences a large symmetry-breaking strain. A quite high (for $BaFe_2As_2$) residual resistivity ratio (RRR ~ 10) is found, attesting for the high quality of our crystals. The inset in Fig. 2a provides more details near $T_N$. Both $\rho_a$ and $\rho_b$ exhibit sharp drops at $T = 138.5$ K, which we identify with the magnetic transition, and $\rho_b$ has a maximum about 5 K above the magnetic transition. The $m_{66}$ of the elastoresistivity tensor has proved very useful for studying the nematic susceptibility $\chi_N$[25–28], can also be calculated for our data since we know the applied anisotropic strain from the thermal expansion of the substrate (Fig. 1b). Here,

$$2m_{66}(T) = \frac{\rho_b(T) - \rho_a(T)}{\frac{1}{2}\left[\rho_b(T) + \rho_a(T)\right]\left[\varepsilon_{\perp}(T) - \varepsilon_{||}(T)\right]}. \quad (1)$$

We find (Fig. 2b) that $|2m_{66}|$ exhibits a very similar magnitude and divergent Curie–Weiss behavior as $T_N$ is approached from above as found in the elastoresistivity data obtained using a piezo-stack[25–28]. This suggests that the resistivity change $\Delta\rho/\rho_0(\varepsilon)$

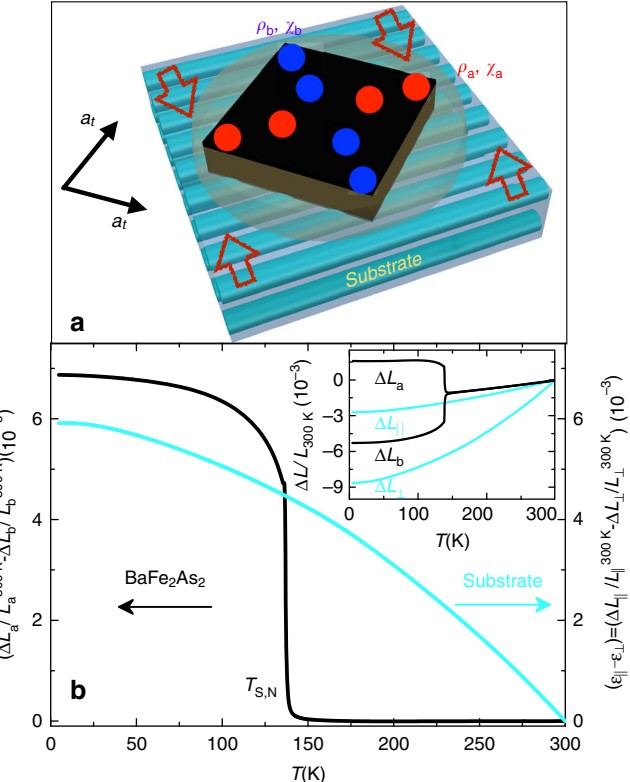

**Fig. 1** Experimental setup. **a** Illustration of the symmetry-breaking straining setup. The crystal is glued on *top* of a glass-fiber-reinforced plastic substrate using epoxy with the $[110]_{tet}$ direction parallel to fibers. Upon cooling, the thermal-expansion anisotropy of the substrate applies a symmetry-breaking strain to the crystal. *Red* and *blue dots* represent electrical contacts along orthorhombic a and b axes (a > b), respectively (see Fig. 2a inset also). **b** Anisotropic strain of the substrate ($L_{\perp}$: perpendicular to fibers, $L_{||}$: parallel to fibers) compared to the in-plane orthorhombic distortion of a free standing $BaFe_2As_2$ crystal ($L_a$: longer orthorhombic axis, $L_b$: shorter orthorhombic axis). The thermal expansion is shown in the inset

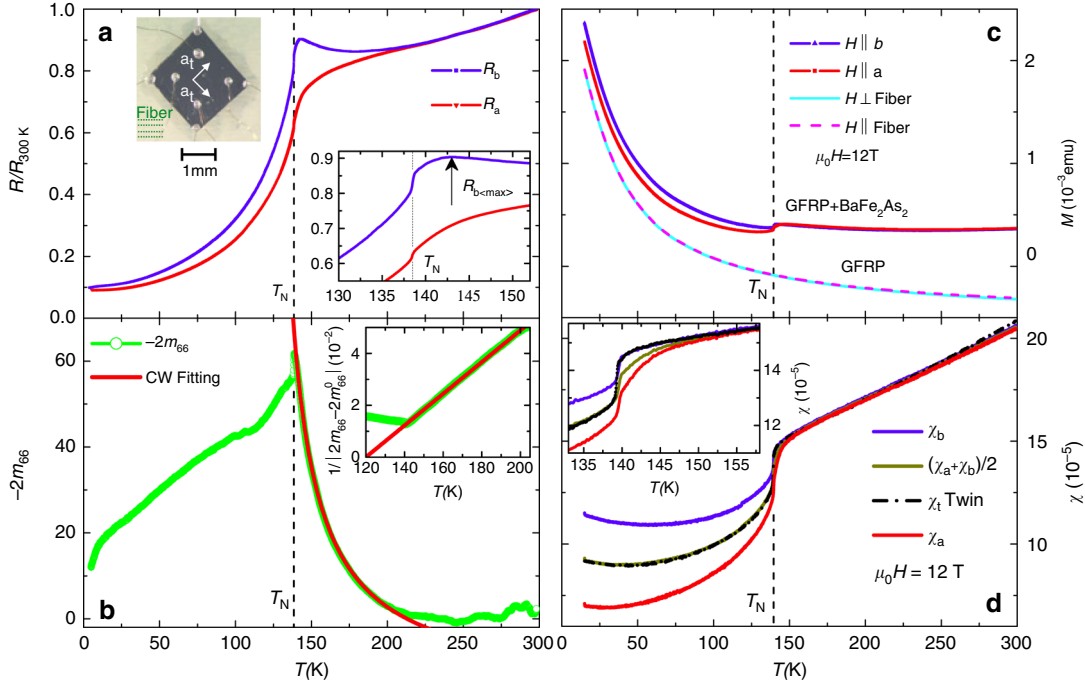

**Fig. 2** In-plane resistance and susceptibility anisotropies. Temperature dependence of **a** the in-plane resistances along $a$ and $b$ directions, **b** the elastoresistivity tensor $2m_{66}$, **c** raw magnetization data of GFRP alone and together with the BaFe$_2$As$_2$ crystal, and **d** anisotropic susceptibility obtained by subtracting the GFRP background from the data shown in **c**. The *red solid line* in **b** is a Curie–Weiss fit (|$2m_{66}$| = $a/(T - T_0)$ + $b$ with $T_0 = 120 \pm 1$ K) and the inset shows the inverse plot. The inset in **a**, **d** displays magnified views near $T_N$. The *arrow* in the inset of **a** indicates a maximum of $R_b$. The photograph in **a** illustrates the contacts configuration of the resistance measurements

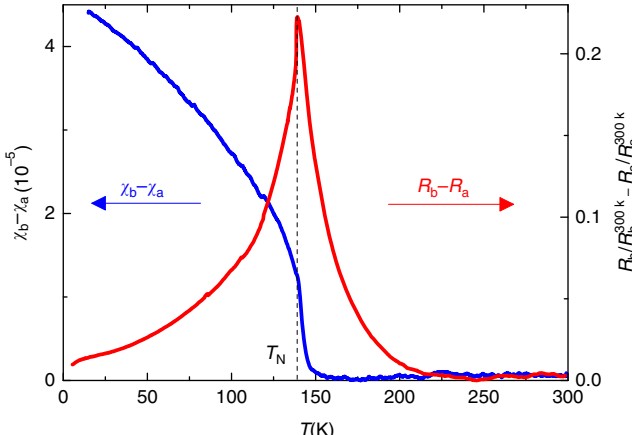

**Fig. 3** Comparison between in-plane resistance anisotropy and susceptibility anisotropy. Temperature dependence of in-plane resistance and susceptibility anisotropies, which demostates that these two anisotropies do not scale with each other

varies approximately linearly with applied strain $\varepsilon$ up to the large strains studied here. We note that the thermal expansion of a typical piezo-stack is also anisotropic and that the elastoresistivity data obtained in these measurements are thus not in the zero-strain limit. Similar to a ferromagnet in an applied field strong enough to nearly saturate the magnetization, we no longer expect a well-defined nematic phase transition for the large strain (on the order of the spontaneous orthorhombic distortion) applied here[46]. We note that this argument only holds if there is a large coupling of the nematic transition to the lattice, which indeed has been verified by elastic[24], Raman scattering[30], and elastoresistivity measurements[28]. Therefore, one expects a very broad transition under application of a large strain, and the observation of a sharp peak in the resistivity anisotropy is quite surprising. Our results possibly suggest that the resistivity anisotropy is more directly related to the magnetic transition than to the nematic fluctuations. We note that a similar conclusion can be deduced from the data of ref. [47], in which the peak in the resistivity anisotropy also occurs at $T_N$ in spite of the fairly large uniaxial pressure applied.

Since the "detwinning apparatus" in our case is reduced to a thin substrate plate, our method is also feasible for investigating the anisotropy of other quantities, e.g., the magnetization. Figure 2c displays the raw magnetization data at 12 Tesla of a BaFe$_2$As$_2$ crystal glued to the glass-fiber substrate in two different field orientations, as well as the bare substrate in the same two orientations. A clear sign of magnetization anisotropy is already observable in the raw data below $T_N$, despite of a considerable Curie–Weiss component in the magnetization of the glass-fiber-reinforced plastic material, which needs to be subtracted. The calculated susceptibility data after subtraction of the substrate background are shown in Fig. 2d along with data of a free-standing crystal in the twinned state. Well above $T_N$, the susceptibilities along both directions are practically identical and decrease linearly with temperature, as previously observed[48] and also exists in other Fe-based systems[49, 50]. Below $T_N$, the susceptibility along the longer axis $\chi_a$ becomes significantly smaller than that of the shorter axis $\chi_b$. The difference between $\chi_a$ and $\chi_b$ keeps increasing with decreasing temperature and the anisotropic ratio $\eta = \chi_b/\chi_a - 1$ reaches ~60% at 15 K. The average of $\chi_a$ and $\chi_b$ agrees excellently with the twinned data $\chi_t$ within the whole temperature range, except slightly above $T_N$ (see inset of Fig. 2d), where the averaged data show a significant precursor to the transition starting at about 150 K. Measurements on a second sample exhibited practically identical behavior, demonstrating the repeatability of this technique (Supplementary Fig. 2). We note that the observed sign, $\chi_b > \chi_a$, explains the sign of the magnetic detwinning effect reported in refs. [51, 52]; however, we observe no anisotropy at ~170 K, as claimed in torque magnetometry experiments on BaFe$_2$As$_2$[53].

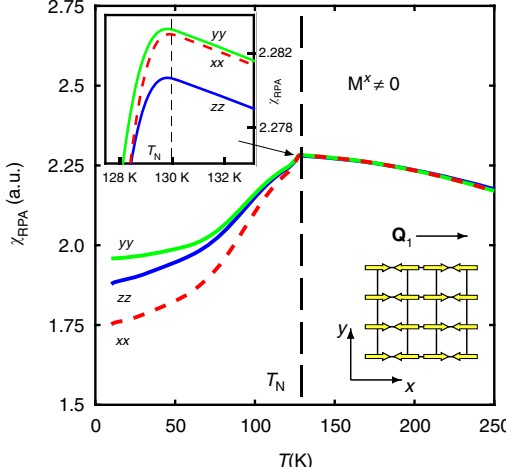

**Fig. 4** Theoretical calculations. Magnetic susceptibility calculated in the stripe AF phase using an itinerant multi-orbital model (coordinate basis is transformed as $a \rightarrow x$, $b \rightarrow y$, $c \rightarrow z$ in comparison with experimental data). The magnetic moments are arranged parallel to the AF wave vector $\mathbf{Q}_1$ so that $M^x \neq 0$, $M^{y,z} = 0$ resulting in $\chi^{yy} > \chi^{xx}$ in agreement with experiment. The inset shows an enlarged view near the transition, where an extremely weak splitting ($\ll 1\%$) between $\chi^{xx}$ and $\chi^{yy}$ occurs in the paramagnetic state due to the finite orbital ordering ($\Delta_{oo} = -25$ meV)

Figure 3 highlights the surprisingly different behavior of the susceptibility anisotropy, $\chi_b - \chi_a$, and the resistivity anisotropy, $\rho_b - \rho_a$. Whereas $\rho_b - \rho_a$ is peaked close to and extends considerably above $T_N$, $\chi_b - \chi_a$ only starts to develop slightly above $T_N$ and then increases to the lowest temperatures. Thus, the resistivity anisotropy and the susceptibility anisotropy do not scale linearly with each other above the transition, in contrary to the expectation of the spin-nematic scenario[10, 11].

**Theoretical calculations**. The most natural way to account for the anisotropy of the magnetic susceptibility in the magnetically ordered state is to include spin–orbit coupling in the effective low-energy model of the iron-based superconductors. Indeed, it is responsible for the observed magnetic anisotropy of the striped AF state, namely, for the alignment of the magnetic moments parallel to the AF wave-vector $\mathbf{Q}_1$ at the transition temperature[54]. We describe (details can be found in Supplementary Note 1) the itinerant electron system of the parent iron-based super-conductors by a multi-orbital Hubbard Hamiltonian, which consists of the non-interacting hopping Hamiltonian within the $3d$-orbital manifold, $H_0$, and Hubbard–Hund interaction, $H_{int}$. We specify the hopping parameters $t_{ij}^{\mu\nu}$ according to the band structure parametrization obtained by Ikeda[55] for a five orbital model or three-orbital model by Daghofer[56, 57] and employ $\lambda \approx$ 40 meV, which is of the same order as found experimentally by angle-resolved photoemission spectroscopy[58]. Besides the band dispersions, the non-interacting Hamiltonian must also contain the spin orbit coupling term $\lambda \mathbf{S} \cdot \mathbf{L}$ with $\mathbf{S}$ and $\mathbf{L}$ denoting the spin and orbital angular momentum operator, respectively. Note that this atomic-like term preserves the Kramers degeneracy of each state. We project this term from the $L = 2$ spherical harmonic basis to the orbital basis using the standard procedure of ref. [54]. In order to simulate the breaking of the $C_4$ symmetry above $T_N$ in the experiment, we also introduced a uniform energy splitting of the $d_{xz}$ and $d_{yz}$ orbitals[59],

$$H_{oo} = \Delta_{oo} \sum_{\mathbf{k}\sigma} \left( c_{xz\mathbf{k}\sigma}^\dagger c_{xz\mathbf{k}\sigma} - c_{yz\mathbf{k}\sigma}^\dagger c_{yz\mathbf{k}\sigma} \right), \qquad (2)$$

where $\Delta_{oo} = -25$ meV (see Supplementary Note 1 for details) was used so that $d_{yz}$ shifts upwards. Note that such a term appears in the striped AF state automatically as a result of the magnetic ordering breaking the $C_4$ symmetry of the lattice.

The results of our susceptibility calculations (see Supplementary Note 1 for details) are shown in Fig. 4. To compare with experimental data, we assign $a \rightarrow x$, $b \rightarrow y$, $c \rightarrow z$. As expected, the sign of in-plane susceptibility anisotropy strongly depends on the orientation of the magnetic moments. Alignment of the magnetic moments along $\mathbf{Q}_1$ driven by spin–orbit coupling produces the anisotropy observed in our magnetization experiments, i.e., $\chi^{yy} > \chi^{xx}$. We note that this is also the same anisotropy expected in a purely localized magnetic model, i.e., the susceptibility is larger for fields perpendicular to the moments. Apart from spin–orbit coupling, the calculation shows that the Umklapp susceptibility dominated by intra-orbital ($yz$, $yz$) contributions is responsible for the observed pronounced anisotropy. We find that spin–orbit coupling plays a crucial role in transferring the anisotropy of the real space and the orbital structure of the electronic wave functions to the iron spins. The striped AF state immediately introduces via spin–orbit coupling the corresponding maximal splitting of the uniform spin susceptibility with the dominant suppression of the $xx$ component of the susceptibility due to the largest magnetic gap in the $yz$ orbital (Supplementary Fig. 1) and translational symmetry breaking allowing for the Umklapp large $\mathbf{Q}$ transfer terms in the uniform susceptibility. The quantitative agreement between the theoretical data and experiment points to a purely magnetic origin of the anisotropy in the spin susceptibility. The inset in Fig. 4 shows that the anisotropy induced by finite orbital ordering in the paramagnetic state is extremely weak $\eta = \chi^{yy}/\chi^{xx} - 1 \ll 1\%$. Orbital ordering therefore can not be responsible for the non-negligible anisotropy slightly above $T_N$ observed in our experimental data. The small effect is however consistent with the comparatively small orbitally induced susceptibility anisotropy in the wide region between 150 and 200 K (Fig. 3).

## Discussion

In summary, we have determined the in-plane susceptibility and resistivity anisotropies of the prototypical parent compound BaFe$_2$As$_2$ using a simple method, which applies a large symmetry breaking in-plane strain. This strain, which is of the same magnitude as the spontaneous orthorhombic distortion below $T_N$, thus breaks the $C_4$ lattice symmetry and thus also lifts the orbital degeneracy of the $d_{xz}$ and $d_{yz}$-derived bands[20]. The observed susceptibility anisotropy in the magnetically ordered phase qualitatively agrees well with calculations using an effective low-energy itinerant model including spin–orbit coupling, in which the sizable splitting is dominated by intra-orbital ($yz$, $yz$) Umklapp processes. Striking is the different behavior of the resistivity and susceptibility anisotropies in the paramagnetic strained state. In particular, whereas the resistivity anisotropy exhibits a Curie–Weiss divergence extending to temperatures much larger than $T_N$, the susceptibility anisotropy develops only about 10 K above $T_N$. Our calculations, in which the strain is simulated by orbital splitting, show that orbital order produces a negligible susceptibility anisotropy above $T_N$ and serve to disentangle anisotropies due to orbital order and lattice distortion from those of the magnetically ordered state. The fact that we still observe a diverging resistivity anisotropy at, or near, the magnetic transition in this highly strained state strongly suggests that pure lattice distortions and/or orbital ordering must play subdominant roles in BaFe$_2$As$_2$, leaving magnetism as the primary player. Finally, we expect that the simple straining technique presented here should be very useful for investigating anisotropies in

other experimental probes, e.g., angle-resolved photoemission spectroscopy, Raman, etc.

## Methods

**Sample growth**. High-quality $BaFe_2As_2$ single crystals were grown from self-flux in alumina crucibles sealed in a steel cylinder in an atmosphere of Ar kept at 1 bar. The cylinder was kept at 1340 °C for 5 h and then cooled down at a very slow rate of 0.11 °C/h. An in situ annealing process was performed during the cooling procedure in which the samples were kept between 850 and 750 °C for more than 800 h.

**Symmetry-breaking straining setup**. To apply large symmetry-breaking strain, samples with typical dimensions of 2 mm × 2 mm × 0.08 mm were glued onto a 3 mm × 3 mm × 0.2 mm glass-fiber-reinforced plastic substrate using two-component epoxy(UHU Plus Endfest 300, 90 min) with the crystal's tetragonal $[110]_{tet}$ direction orientated parallel to the fibers (Fig. 1a).

**Strain determination**. In order to determine the anisotropic strain applied to the sample, the thermal expansion of the glass-fiber-reinforced plastic substrate material was characterized by a home-built high-resolution capacitance dilatometer[60].

**Resistance and magnetization measurements**. Electrical contacts, with typical resistances of around $2\Omega$, were made using silver paste, and the sample resistances along two perpendicular directions were measured simultaneously on the same sample by a four-terminal method (see Fig. 1a or Fig. 2a inset). Magnetization measurements both parallel and perpendicular to the fiber orientation of the substrate were carried out in a Physical Property Measurement System (PPMS) using the Vibrating Sample Magnetometer (VSM) unit from Quantum Design Inc.

**Data availability**. The data that support the current findings are available from the corresponding authors upon reasonable request.

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

## Acknowledgements

We thank Rafael Fernandes and Igor Mazin for valuable discussions. We acknowledge support by Deutsche Forschungsgemeinschaft and Open Access Publishing Fund of Karlsruhe Institute of Technology. I.E. acknowledges support by the Ministry of Education and Science of the Russian Federation in the framework of Increase Competitiveness Program of NUST MISiS (K2-2016-067).

## Author contributions

C.M. initiated and supervised this study. M.H. performed the resistivity and magnetization experiments. L.W. conducted the thermal expansion measurements. T.W. and P.A. synthesized the samples. M.H. and T.W. prepared the glass-fiber-reinforced plastic (GFRP) substrate. F.A. and I.E. performed the theoretical calculations. M.H., F.H., I.E., J. S., and C.M. analyzed the data and wrote the manuscript with contributions from all other authors.

## Additional information

**Competing interests:** The authors declare no competing financial interests.

