## [Peer Review File · Nature Communications]

Reviewers' Comments:

Reviewer #1 (Remarks to the Author):

The authors present a study of BaFe₂As₂ crystal(s) adhered to a substrate with a very anisotropic thermal contraction. Upon cooling, the substrate applies a strong, controlled anisotropic strain to the sample. Overall, it is a simple and promising technique, and the authors obtain an important result, that the resistivity and susceptibility anisotropies have very different temperature dependences.

The first (minor) point I would like to make is that the strain applied to the sample is almost certainly not uniaxial, because neither the lattice constant along x or y is free to vary according to the thermal expansion of BaFe₂As₂: the substrate constrains both. It would be more accurate to describe the applied strain as anisotropic than as uniaxial.

A more substantial point concerns the interpretation of previous results, based on piezoelectric actuators. In previous studies, piezoelectric actuators have been used to apply small, tunable strains to iron-based samples adhered to their surfaces, yielding, above the nematic transition, diverging elastoresistivities that have been interpreted as evidence that the nematic transition is electronically driven. In this work, the authors propose that because they see the same divergence with a much larger strain that will smear out any purely nematic transition, the resistivity anisotropy is related to magnetic rather than nematic order. This particular argument is technically flawed, however, because piezoelectric actuators also have strongly anisotropic thermal contractions. This fact has not been appreciated in the literature, but it means that piezo-based elastoresistivity results are not in fact, as the authors write in this manuscript, derived from measurements in the zero-strain limit. Data sheets from Physik Instrumente (a manufacturer of piezoelectric actuators), for example, indicate an expansion of ~0.1% along the length of an actuator as it is cooled to cryogenic temperatures, and a contraction of 0.1-0.2% in the transverse directions. This is not as strong an absolute differential as the authors' GFRP substrate, but it is only ~50% smaller, not an order of magnitude smaller as the authors write.

The authors' argument, that diverging elastoresistance even in the presence of strong anisotropic strain means that the resistivity is driven by magnetic order, is also scientifically inadequate. The argument that diverging elastoresistivity indicates diverging nematic susceptibility is based on an assumed linear relationship between the strength of the applied symmetry-breaking field and resistivity anisotropy, and also between the resistivity anisotropy and the amplitude of a nematic

order. In this work, the authors show (and state that they have shown) that the former linear relationship extends to large strains. That fact does not invalidate the argument that it is related to a diverging nematic susceptibility. The fact that the nematic transition itself will be smeared out by the large applied strain might not be important: it might affect the resistivity only in a small region around the erstwhile nematic transition, just like a ferromagnet in a weak magnetic field. I do not mean to insist that the previous interpretation is correct, only to say that this particular argument of the authors is not adequate.

I believe that the authors are on firmer ground when arguing from the observation that resistivity anisotropy falls sharply below the Neel transition. But the authors do not argue from a specific model of the origin of the transport anisotropy, a topic that has been fairly extensively studied in the iron-based superconductors. Also, the authors contradict themselves later in the manuscript: The main point of this manuscript is that the susceptibility and resistivity anisotropies are very different. The authors argue firmly, and with solid theoretical backing, that the susceptibility anisotropy is due to magnetic order, and propose that the different behaviors of resistivity and susceptibility anisotropy allow magnetic and nematic orders to be disentangled. Meaning that resistivity is a probe of the nematic order?

I have four further experimental points, three minor and one major. First, the authors argue that the low shear modulus of BaFe₂As₂ in the vicinity of the structural transition ensures good strain transmission. What about above and below the transition? The second point is that the authors state they measured ρ_a and ρ_b on the same sample. Using a van der Pauw contact configuration? Third, is this measurement the first direct measurement of the susceptibility anisotropy of a detwinned iron pnictide sample? This is not clearly stated in the manuscript. And finally, my major point: How many samples did the authors measure? There is no indication in the main text that key results were verified through measurement of a second sample.

This review is rather negative so I do want to emphasize that overall I believe the authors' technique to be promising, and that the different forms of the resistivity and susceptibility anisotropies is a substantial result. This particular manuscript however should be more tightly argued in regards to the resistivity anisotropy, however, and key results verified through measurement of a second sample.

Reviewer #2 (Remarks to the Author):

In reviewing the manuscript "Dichotomy between the in-plane magnetic susceptibility and resistivity anisotropies between extremely strained

BaFe₂As₂" by He et al., what at first glance appeared to be yet another study of the transport anisotropy in this material quickly turned to an appreciation of an important technical achievement in the area of creating a twin-free crystal below the structural and magnetic transitions in this material, as well as the interesting physics that flows from this. The large (uniaxial) strain leads to interesting effects; a large resistivity anisotropy is observed above and below the magnetic transition, whereas the susceptibility anisotropy is observed mainly below the transition. This behavior suggests that the resistivity anisotropy appears to be more a result of the magnetic transition than nematic fluctuations. Based on this result, as well as the novel experimental approach, I recommend that this work be published in Nature Communications. I have a couple of minor comments that I would like to the authors to address prior to the publication of this work:

page 1, second column, around line 8: I am confused - in this material, the magnetic and structural transitions are degenerate, as the authors have indicated in Fig. 1(b). It is only in the doped materials that there is a clear separation of the structural and magnetic transitions. If these two transitions are not degenerate in this material, then the authors should indicate explicitly that this is the case and state clearly what the two temperatures are.

page 2, column 1, second paragraph: while the METHODS section provides details about the sample size, is the substrate considerably larger than the sample? How thick is the substrate? Was the substrate prepared especially for this experiment, or was it modified from commercially-available stock? Later in the paragraph, when discussing the magnetic transition at 138.5 K; it would be nice if Fig. 2 had dotted lines for this transition in all four panels to more easily allow the reader to see the range over which fluctuations occur above this temperature. I might add in passing that Fig. 2(d) is a really nice result.

Reviewer #3 (Remarks to the Author):

Mingquan He et. al report in-plane resistivity and uniform magnetic susceptibility anisotropies of the parent compound BaFe₂As₂ of Fe-based superconductor applying a large symmetry-breaking uniaxial strain using a substrate with a very anisotropic thermal expansion. They found large resistivity anisotropy above Neel temperature T_N , in contrast, the anisotropy of the in-

plane magnetic susceptibility develops largely (only) below T_N . Using an itinerant model calculation, the authors show that the observed anisotropy ($\chi_b > \chi_a$) is determined by spin-orbit coupling and the orientation of the magnetic moments in the antiferromagnetic phase, and that the anisotropy is dominated by intra-orbital (yz, yz) contributions of the Umklapp susceptibility.

The authors explained the observed nonlinearity between susceptibility anisotropy and resistivity anisotropy above the magnetic transition: “the susceptibility anisotropy is due to the combination of nematic/orbital order and spin-orbit coupling is much weaker than the one caused by the anisotropy due to long range magnetic order”, claiming the combination of susceptibility and resistivity anisotropy as a tool to disentangle the above two phenomena.

This manuscript contain good scientific work, however, it would be more suitable to a more specialized journal. It will not be able to draw broad interest of the readers of nature communications, therefore, I do not recommend publication in Nature Communications. Nature Scientific report may be considered.

Response to Reviewer #1

“The authors present a study of BaFe₂As₂ crystal(s) adhered to a substrate with a very anisotropic thermal contraction. Upon cooling, the substrate applies a strong, controlled anisotropic strain to the sample. Overall, it is a simple and promising technique, and the authors obtain an important result, that the resistivity and susceptibility anisotropies have very different temperature dependences.”

We appreciate the Referee’s overall positive evaluation of our work. Please find below our detailed response to the comments, which we have tried to incorporate into the revised manuscript as best as possible.

“The first (minor) point I would like to make is that the strain applied to the sample is almost certainly not uniaxial, because neither the lattice constant along x or y is free to vary according to the thermal expansion of BaFe₂As₂: the substrate constrains both. It would be more accurate to describe the applied strain as anisotropic than as uniaxial.”

We totally agree with the reviewer and have changed ‘uniaxial’ to ‘symmetry-breaking strain’ in the manuscript.

“A more substantial point concerns the interpretation of previous results, based on piezoelectric actuators. In previous studies, piezoelectric actuators have been used to apply small, tunable strains to iron-based samples adhered to their surfaces, yielding, above the nematic transition, diverging elastoresistivities that have been interpreted as evidence that the nematic transition is electronically driven. In this work, the authors propose that because they see the same divergence with a much larger strain that will smear out any purely nematic transition, the resistivity anisotropy is related to magnetic rather than nematic order. This particular argument is technically flawed, however, because piezoelectric actuators also have strongly anisotropic thermal contractions. This fact has not been appreciated in the literature, but it means that piezo-based elastoresistivity results are not in fact, as the authors write in this manuscript, derived from measurements in the zero-strain limit. Data sheets from Physik Instrumente (a manufacturer of piezoelectric actuators), for example, indicate an expansion of ~0.1% along the length of an actuator as it is cooled to cryogenic temperatures, and a contraction of 0.1-0.2% in the transverse directions. This is not as strong an absolute differential as the authors’ GFRP substrate, but it is only ~50% smaller, not an order of magnitude smaller as the authors write.”

We thank the referee for pointing out that the piezo-actuators also have a large anisotropic thermal expansion - a fact we were not aware of. Indeed, the differential thermal expansion between the piezoelectric actuator and the sample applies a substantial anisotropic strain to the crystals, as can be inferred by a careful reading of two recent articles from Ian Fisher’s group (M. C. Shapiro et al, Review of Scientific Instruments **87**, 063902 (2016); H-H Kuo et al, Science, **352**, 958 (2016)). We now realise that the zero-strain state may not be reached within the tuneable range of the piezo-actuators, especially at low temperatures. The strain applied by the piezostack can be as large as $\sim 1.5 \times 10^{-3}$ near the transition (see Figure S4 of the supplementary material for H-H Kuo et al, Science, **352**, 958 (2016)). Our previous statement that the nematic susceptibility probed by piezostack experiments is in the zero-strain limit which is 10 times smaller than the strain applied by GFRP substrate was thus incorrect. We have removed all references to “zero strain limit” in the text and mention in a footnote that the piezostack also applies an appreciable symmetry breaking strain due to its anisotropic thermal expansion.

On the other hand, we would still like to stick to our argument that it is quite surprising that we still observe a nearly diverging resistivity anisotropy (and associated nematic susceptibility) for the large strains applied in our experiment, and that this seems to imply a magnetic origin (see reasoning below).

“The authors' argument, that diverging elastoresistance even in the presence of strong anisotropic strain means that the resistivity is driven by magnetic order, is also scientifically inadequate. The argument that diverging elastoresistivity indicates diverging nematic susceptibility is based on an assumed linear relationship between the strength of the applied symmetry-breaking field and resistivity anisotropy, and also between the resistivity anisotropy and the amplitude of a nematic order. In this work, the authors show (and state that they have shown) that the former linear relationship extends to large strains. That fact does not invalidate the argument that it is related to a diverging nematic susceptibility. The fact that the nematic transition itself will be smeared out by the large applied strain might not be important: it might affect the resistivity only in a small region around the erstwhile nematic transition, just like a ferromagnet in a weak magnetic field. I do not mean to insist that the previous interpretation is correct, only to say that this particular argument of the authors is not adequate.”

Yes, we agree that our argument depends strongly on how large the smearing of the transition is. Taking the example of a ferromagnet, a small magnetic field will only lead to broadening close to T_c . On the other hand, a field large enough to practically fully polarize the spins will create a T-broadening on the order of the critical temperature itself - i.e. a huge broadening. We believe that the large strain applied by our substrate puts BaFe₂As₂ into the latter category, because the symmetry-breaking strain applied is on the same order as the spontaneous orthorhombic distortion. It is thus a quite large field in the magnetic language.

We have modified the text to more clearly present the above argumentation. On the other hand, we have also placed less emphasis on this point by e.g. removing it from the abstract.

We also note that other experimental results confirm also confirm our experimental findings. H. Man et al, PRB **92**, 134521 (2015) (Ref. 45 in the manuscript) found that the resistivity anisotropy diverges at T_N although a large uni-axial stress was applied.

“I believe that the authors are on firmer ground when arguing from the observation that resistivity anisotropy falls sharply below the Neel transition. But the authors do not argue from a specific model of the origin of the transport anisotropy, a topic that has been fairly extensively studied in the iron-based superconductors. Also, the authors contradict themselves later in the manuscript: The main point of this manuscript is that the susceptibility and resistivity anisotropies are very different. The authors argue firmly, and with solid theoretical backing, that the susceptibility anisotropy is due to magnetic order, and propose that the different behaviors of resistivity and susceptibility anisotropy allow magnetic and nematic orders to be disentangled. Meaning that resistivity is a probe of the nematic order?”

It is true that we did not propose a specific model for the resistivity anisotropy and we only report the experimental findings here. The main point of our work was to study the susceptibility anisotropy, and we use the resistivity to demonstrate that our straining technique works effectively. We fully agree that more systematic experiments and theoretical works are necessary to capture the universal (if exist) microscopic origin of the resistivity anisotropy in Fe-based materials.

We are sorry for the confusion made in our previous text concerning the disentanglement between magnetic order and orbital/nematic order. The reviewer is right - our results do not allow us to disentangle magnetic from nematic order, and this sentence has been removed.

On the other hand, our experiment and theoretical considerations allow us to place pure lattice effects, as well as orbital order (which is induced by the strain), on the lower side of the interaction hierarchy, as hopefully now explained in the present text.

“I have four further experimental points, three minor and one major. First, the authors argue that the low shear modulus of BaFe₂As₂ in the vicinity of the structural transition ensures good strain transmission. What about above and below the transition? The second point is that the authors state they measured ρ_a and ρ_b on the same sample. Using a van der Pauw contact configuration? Third, is this measurement the first direct measurement of the susceptibility anisotropy of a detwinned iron pnictide sample? This is not clearly stated in the manuscript. And

finally, my major point: How many samples did the authors measure? There is no indication in the main text that key results were verified through measurement of a second sample.”

We thank the Referee for these useful comments. Concerning the experimental aspects.

1. As shown in Fig. 2a,b, almost no resistivity anisotropy can be resolved above ~ 220 K, which means that strain is not well transmitted above ~ 220 K. This is also seen in the piezostack experiments: “ $T = 250$ K (the highest temperature at which strain can be effectively transmitted by the epoxy) ” by H-H Kuo et al, Science, **352**, 958 (2016). Below the transition, BaFe₂As₂ spontaneously distorts and the strain will act to detwin the crystal. The softening of the shear modulus stops sharply below T_N and the modulus is a constant below T_N . (A. E. Böhmer et al, PRL **112**, 047001 (2014)). Therefore, the strain transmission below the transition should be no problem.
2. The resistivities ρ_a and ρ_b were measured simultaneously on the same sample with the contacts configuration shown in the revised Fig.1a.
3. Yes, this is the first direct measurement of susceptibility anisotropy for a detwinned iron pnictide. We have incorporated this into the revised version.
4. We have repeated resistivity and susceptibility measurements on 3 different samples. Results of the susceptibility anisotropy of a second sample are shown in the supplementary Figure 2.

Response to Reviewer #2

"In reviewing the manuscript "Dichotomy between the in-plane magnetic susceptibility and resistivity anisotropies between extremely strained BaFe₂As₂" by He et al., what at first glance appeared to be yet another study of the transport anisotropy in this material quickly turned to an appreciation of an important technical achievement in the area of creating a twin-free crystal below the structural and magnetic transitions in this material, as well as the interesting physics that flows from this. The large (uniaxial) strain leads to interesting effects; a large resistivity anisotropy is observed above and below the magnetic transition, whereas the susceptibility anisotropy is observed mainly below the transition. This behavior suggests that the resistivity anisotropy appears to be more a result of the magnetic transition than nematic fluctuations. Based on this result, as well as the novel experimental approach, I recommend that this work be published in Nature Communications."

We thank the referee for the encouraging and positive evaluations. We have revised our manuscript according to the referee's remarks as listed below.

"page 1, second column, around line 8: I am confused - in this material, the magnetic and structural transitions are degenerate, as the authors have indicated in Fig. 1(b). It is only in the doped materials that there is a clear separation of the structural and magnetic transitions. If these two transitions are not degenerate in this material, then the authors should indicate explicitly that this is the case and state clearly what the two temperatures are."

We are sorry for the confusion. In pure BaFe₂As₂, magnetic transition and structural transition coincide with each other. In Fig. 1b, the thermal expansion of BaFe₂As₂ was measured on a free standing sample in which $T_N = T_S$. In the resistivity and magnetic susceptibility measurements, the structural transition is smeared out by the large symmetry-breaking strain applied from the substrate. Therefore, the structural transition is no longer well defined, but the magnetic transition remains intact. Hence, only T_N is marked in Fig. 2, Fig. 3 and Fig. 4.

"page 2, column 1, second paragraph: while the METHODS section provides details about the sample size, is the substrate considerably larger than the sample? How thick is the substrate? Was the substrate prepared especially for this experiment, or was it modified from commercially-available stock?"

The relation of the substrate size to the sample size is roughly as shown in Fig. 1a, with a thickness of 0.2 mm. It was cut from a larger stock of a commercial product. This information was added to the 'Experimental section' of the paper.

"Later in the paragraph, when discussing the magnetic transition at 138.5 K; it would be nice if Fig. 2 had dotted lines for this transition in all four panels to more easily allow the reader to see the range over which fluctuations occur above this temperature. I might add in passing that Fig. 2(d) is a really nice result."

We thank the referee for this nice remark. We have added T_N lines for all panels in Fig. 2

Response to Reviewer #3

Reviewer #3 (Remarks to the Author):

“Mingquan He et. al report in-plane resistivity and uniform magnetic susceptibility anisotropies of the parent compound BaFe_2As_2 of Fe-based superconductor applying a large symmetry-breaking uniaxial strain using a substrate with a very anisotropic thermal expansion. They found large resistivity anisotropy above Neel temperature T_N , in contrast, the anisotropy of the in-plane magnetic susceptibility develops largely (only) below T_N . Using an itinerant model calculation, the authors show that the observed anisotropy ($\chi_b > \chi_a$) is determined by spin-orbit coupling and the orientation of the magnetic moments in the antiferromagnetic phase, and that the anisotropy is dominated by intra-orbital (yz, yz) contributions of the Umklapp susceptibility.

The authors explained the observed nonlinearity between susceptibility anisotropy and resistivity anisotropy above the magnetic transition: “the susceptibility anisotropy is due to the combination of nematic/orbital order and spin-orbit coupling is much weaker than the one caused by the anisotropy due to long range magnetic order”, claiming the combination of susceptibility and resistivity anisotropy as a tool to disentangle the above two phenomena.

This manuscript contain good scientific work, however, it would be more suitable to a more specialized journal. It will not be able to draw broad interest of the readers of nature communications, therefore, I do not recommend publication in Nature Communications. Nature Scientific report may be considered.”

We thank the referee for reviewing our work.

List of Changes

1. The Abstract was revised to better fit the style of Nature Communications.
2. Two more references (Ref.20 and 21) were added to the Introduction section.
3. Brief titles of each Figure and each Section were added.
4. Figure 1a was modified to better illustrate the resistivity measurements, and the description of Figure 1a was updated accordingly.
5. Figure 2 was updated with the magnetic transition marked by vertical lines in all panels, as suggested by Reviewer 2.
6. The brief summary part in the introduction section was revised—line 41-49.
7. The term 'uniaxial strain' was modified to 'symmetry-breaking strain' throughout the text, following the suggestion of Reviewer 1.
8. A footnote was added in page 2 concerning the non-negligible anisotropic strain of piezo-stack according to the comments of Reviewer 1.
9. The interpretation of the elastoresistivity results was updated—line 99-106.
10. Susceptibility measurements of a second sample was included in the Supplementary Fig.2, which is also described in line 141-143 in the main text according to the suggestion of Reviewer 1.
11. The conclusion was reformulated—line 229-236, line 252-260.
12. The Methods section was revised (line 272-274, line 288) according to suggestions from Reviewer 1 and Reviewer 2.

Reviewers' Comments:

Reviewer #1 (Remarks to the Author):

The revised manuscript is considerably improved. I still have comments however that I feel should be addressed before this manuscript is published.

(1) A minor point: in the first paragraph of the results section, the authors state, "a symmetry-breaking strain on the order of $\sim 4 \times 10^{-3}$." It would be helpful if the authors stated precisely what quantity is 4×10^{-3} ; I expect it is $e_{xx} - e_{yy}$, where e_{xx} is the strain along the x axis. Also, the \sim symbol means "on the order of," so writing this in words too is redundant.

(2) Important to the authors' technique is not only that the substrate is not too large compared with the sample, but also that it is not strongly magnetic.

(3) Another minor point: though not essential, I think that a photograph of a mounted sample would be helpful, so the reader can judge how well the authors have really achieved the contact configuration they state.

(4) A more serious technical point. With the authors' contact configuration, the current flow will not be uniformly along either the a or the b axes. Therefore, the measured resistance (not resistivity) R_a will depend on both ρ_a and ρ_b , and likewise for R_b . If the contact configuration is known with sufficient precision, meaning that the contacts are sufficiently small in comparison with the sample size, it may be possible to simulate the current flow within the sample and extract ρ_a and ρ_b from the measured R_a and R_b . It is not in principle sufficient to divide by the 300 K measurements, because ρ_a and ρ_b have separate temperature dependences. I think that the authors have two choices: one is to perform the necessary analysis to extract ρ_a and ρ_b from R_a and R_b , and the other is to appropriately change the language in the text and the labels in the figures to reflect the fact that they are measuring and plotting resistances, not resistivities. With the latter approach, I believe that the authors could argue that the divergence in resistivities will be quantitatively similar to the observed divergence in resistances, and their discussion would still stand.

(5) On a similar note, the authors are measuring resistances with contacts placed on top of a layered sample. The resistive anisotropy ρ_c/ρ_a therefore needs to be mentioned; if I recall correctly, the c/a resistivity anisotropy of BaFe₂As₂ is not huge, so it is likely that in the authors' setup the current flow is largely homogeneous along the c axis. However this point needs to be addressed in the manuscript.

(6) A scientific point. The authors state in their response to my previous comments and in the manuscript that because the strain they apply is comparable to the structural distortion that occurs in a free sample, the nematic transition must be dramatically broadened. This argument is incomplete, because it neglects the coupling between a nematic order parameter and the lattice. If this coupling is very small, then a "strong" nematic order might induce only a "small" lattice distortion, and applying a strain comparable to this unconstrained-sample distortion would not dramatically broaden the nematic transition. The authors' conclusion that orbital order plays a subdominant role in determining the resistivity anisotropy is therefore incomplete.

(7) Why did the authors choose $\Delta_{oo} = 25$ meV in their simulation?

Reviewer #2 (Remarks to the Author):

I have reviewed the revised manuscript on the "Dichotomy between in-plane magnetic susceptibility and resistivity anisotropies in extremely strained BaFe₂As₂" by He et al., and find that the authors have addressed all of my previous concerns. Moreover, I have also read the response to the comments of Reviewer #1, and I am satisfied that they have done a good job in addressing their points (although I can not speak for the Reviewer in this case). In terms of suitability as a Nature Communication, I think that emphasis has to be placed not only on results but also on technique. This is a simple approach for creating a twin-free orthorhombic sample below the Neel transition - this is an important experimental approach that may be easily extended to other techniques, allowing new insights into this material. On this basis, I think that this work will be of interest to a wide audience and I would recommend publication as a Nature Communication.

Response to Reviewer #1

“(1) A minor point: in the first paragraph of the results section, the authors state, "a symmetry-breaking strain on the order of $\sim 4 \times 10^{-3}$." It would be helpful if the authors stated precisely what quantity is 4×10^{-3} ; I expect it is $e_{xx} - e_{yy}$, where e_{xx} is the strain along the x axis. Also, the \sim symbol means "on the order of," so writing this in words too is redundant.”

We have modified the text to ‘a symmetry-breaking strain $\epsilon_a - \epsilon_b \sim 4 \times 10^{-3}$ ’.

“(2) Important to the authors' technique is not only that the substrate is not too large compared with the sample, but also that it is not strongly magnetic.”

We thank the referee for this useful comment, which we have incorporated into the revised manuscript.

“(3) Another minor point: though not essential, I think that a photograph of a mounted sample would be helpful, so the reader can judge how well the authors have really achieved the contact configuration they state.”

We have added a photo of the mounted sample for resistivity measurements as an inset of Fig 2a.

“(4) A more serious technical point. With the authors' contact configuration, the current flow will not be uniformly along either the a or the b axes. Therefore, the measured resistance (not resistivity) R_a will depend on both ρ_a and ρ_b , and likewise for R_b . If the contact configuration is known with sufficient precision, meaning that the contacts are sufficiently small in comparison with the sample size, it may be possible to simulate the current flow within the sample and extract ρ_a and ρ_b from the measured R_a and R_b . It is not in principle sufficient to divide by the 300 K measurements, because ρ_a and ρ_b have separate temperature dependences. I think that the authors have two choices: one is to perform the necessary analysis to extract ρ_a and ρ_b from R_a and R_b , and the other is to appropriately change the language in the text and the labels in the figures to reflect the fact that they are measuring and plotting resistances, not resistivities. With the latter approach, I believe that the authors could argue that the divergence in resistivities will be quantitatively similar to the observed divergence in resistances, and their discussion would still stand.”

We agree with the Referee that we are strictly measuring resistance and not directly the anisotropic resistivities. For small ratios of the anisotropies as in our case (roughly 1.3), the error is, however, expected to be small (see e.g. I. Miccoli, J. Phys.: Condens. Matter **27** (2015) 223201). For the sake of correctness, we have therefore changed the labels of the Figures from resistivity to resistance. In the text we now argue that the resistivity anisotropy is closely related to the resistance anisotropy for our contact scheme if the anisotropy is small.

“(5) On a similar note, the authors are measuring resistances with contacts placed on top of a layered sample. The resistive anisotropy ρ_c/ρ_a therefore needs to be mentioned; if I recall correctly, the c/a resistivity anisotropy of BaFe2As2 is not huge, so it is likely that in the authors' setup the current flow is largely homogeneous along the c-axis. However this point needs to be addressed in the manuscript.”

The ratio $\gamma = \rho_c/\rho_a$ varies from 3 to 6 across the whole temperature range [M. A. Tanatar et al, Phys. Rev. B **79**, 134528(2009)]. We have added this reference and incorporated this point into the the manuscript.

“(6) A scientific point. The authors state in their response to my previous comments and in the manuscript that because the strain they apply is comparable to the structural distortion that occurs

in a free sample, the nematic transition must be dramatically broadened. This argument is incomplete, because it neglects the coupling between a nematic order parameter and the lattice. If this coupling is very small, then a "strong" nematic order might induce only a "small" lattice distortion, and applying a strain comparable to this unconstrained-sample distortion would not dramatically broaden the nematic transition. The authors' conclusion that orbital order plays a subdominant role in determining the resistivity anisotropy is therefore incomplete."

We thank the Referee for this comment, which helps us to state our conclusions more precisely. We totally agree with the Referee that the degree of broadening is expected to depend strongly on the magnitude of the nematic-lattice coupling. The coupling of the SDW or nematic transition to the lattice has indeed been shown to be quite strong in Ba122, as evidenced by the large lattice-coupling induced shift of the transition temperature (up to 40 K) extracted by analyzing the temperature dependence of the shear modulus softening (A. E. Böhmer and C. Meingast, *Comptes Rendus Physique* **17**, 90 (2016)), from Raman scattering data (Y. Gallais and I. Paul, *Comptes Rendus Physique* **17**, 113 (2016)) or from elastoresistivity measurements (~30 K enhancement (H.-H. Kuo et al, *Science* **352**, 958 (2016))). This shift amounts to almost a 30 % increase of the transition temperature. We therefore conclude that the coupling is not small and that a strain of the order of the spontaneous lattice distortion would result in significant broadening of the nematic transition. We have added a sentence discussing the above and hope that our argument is now more complete.

"(7) Why did the authors choose $\Delta_{oo} = 25$ meV in their simulation?"

The magnitude of the orbital order has not been chosen arbitrarily. In particular, in the striped antiferromagnetic state with $(\pi,0)$ ordering wave-vector this value of Δ_{oo} appears naturally as a result of the self-consistent mean-field calculations of the dominant magnetic order due to lowering the crystal symmetry of the lattice from the tetragonal (C4) to the orthorhombic one (C2) in the antiferromagnetic state. The degeneracy between the xz and yz orbitals is lifted at the Gamma-point in the C2 phase, which yields for the parameters of the model $\Delta_{oo}=25$ meV.

To test the effect of the orbital nematic order on the χ_0 alone, i.e. without a magnetic order, we simply set magnetic order to zero but keep Δ_{oo} as before. Obviously the results demonstrate that in ferropnictides the Δ_{oo} alone cannot explain the experimental data without invoking magnetic degrees of freedom. Also, this value of Δ_{oo} matches that observed by ARPES within a factor of two [M. Yi, *PNAS* **108**, 6878 (2011)].

List of Changes

Changes in the main text are marked in red.

1. We specified the symmetry-breaking strain in Line 61-63 as suggested by reviewer 1.
2. One more advantage of our technique was added in to the main text in Line 72-74.
3. A photograph of the resistance measurements set-up was updated into the Fig.2a inset.
4. The resistivity labels in Fig.2a and Fig 3 were modified to resistance according to the comments of reviewer 1.
5. The validity of relating the resistance anisotropy to resistivity anisotropy was added in the main text in Line 80-86.
6. A short discussion about the nematic-lattice coupling strength was incorporated (Line 113-120), which makes our augment more complete.

Reviewer #1 (Remarks to the Author):

The authors have satisfied my concerns over this manuscript, and I now recommend publication.

On line 42, there is a repeated "and."

The photograph shows that the authors applied their electrical contacts very neatly. The current contacts are closer to the corners than I thought from their diagram. I point out that, using corner contacts alone, the authors could in future do a Montgomery method measurement to measure ρ_{xx} and ρ_{yy} of a sample simultaneously and in a mathematically accurate way.

I appreciate that the authors have addressed the nematicity-lattice coupling. The authors' argument relies on a quantitative statement, that the strain-induced transition broadening is substantial on the scale of T_S , that is, at least a few 10 K. Their discussion is qualitative and does not fully support this statement, however on the basis of available data it is adequate.

If there is space in the manuscript, a sentence justifying their choice of $\Delta_{oo} = 25$ meV would be helpful.

Overall, I recommend publication and I hope this article is well received.

Response to Reviewer #1

“The authors have satisfied my concerns over this manuscript, and I now recommend publication. Overall, I recommend publication and I hope this article is well received.”

We thank the referee for his/her useful critique, which helped to improve our manuscript.

“On line 42, there is a repeated “and.””

We have corrected this typo mistake.

“The photograph shows that the authors applied their electrical contacts very neatly. The current contacts are closer to the corners than I thought from their diagram. I point out that, using corner contacts alone, the authors could in future do a Montgomery method measurement to measure ρ_{xx} and ρ_{yy} of a sample simultaneously and in a mathematically accurate way.”

We thank the referee for the nice suggestion. We are planning the Montgomery measurements in the near future.

“I appreciate that the authors have addressed the nematicity-lattice coupling. The authors' argument relies on a quantitative statement, that the strain-induced transition broadening is substantial on the scale of T_S , that is, at least a few 10 K. Their discussion is qualitative and does not fully support this statement, however on the basis of available data it is adequate.”

“If there is space in the manuscript, a sentence justifying their choice of $\Delta_{00} = 25$ meV would be helpful.”

We have added few sentence explaining how the value of Δ_{00} was chosen in the Supplementary Note 1.